# Variations in face experiences during the COVID-19 pandemic affect infants' preference for their mother's face

Megumi Kobayashi[1]*, Machi Sugai[2], So Kanazawa[3], Masami K. Yamaguchi[2]

1 Department of Psychology, Faculty of Humanities, Niigata University, Niigata, Japan, 2 Department of Psychology, Faculty of Letters, Chuo University, Tokyo, Japan, 3 Department of Psychology, Faculty of Integrated Arts and Social Sciences, Japan Women's University, Tokyo, Japan

* mkobayashi@human.niigata-u.ac.jp

## Abstract

The Coronavirus disease 2019 pandemic has significantly altered infants' social environments. In Japan, people continued to wear face masks outdoors for approximately three years following the onset of the pandemic. Consequently, most infants born during this period have limited exposure to unmasked faces. However, how such changes in face experience may influence the development of face processing during infancy remains unclear. In the present study, we investigated visual preferences for mothers' faces versus unfamiliar female faces in 5–8-month-old infants born during the pandemic in three regions of Japan (Tokyo in Experiment 1 and provincial cities in Experiment 2). Infants' preferences for their mothers' faces were evaluated under two conditions: when both faces were unmasked (unmasked condition) and when both wore face masks (masked condition). Additionally, parents completed questionnaires reporting their infant's exposure to unmasked faces in daily life. Group-level analysis showed that infants in provincial cities significantly preferred their mothers' faces under both masked and unmasked conditions. By contrast, infants in Tokyo showed a significant preference for their mothers' faces only in the masked condition. Moreover, individual-level analyses revealed that infants with greater exposure to unmasked faces were more likely to prefer their mothers in the unmasked condition. These findings suggest that infants' daily visual experiences, particularly with unmasked faces, may modulate their developing preferences for their mothers' faces, highlighting the impact of social context on early face processing.

## Introduction

The Coronavirus disease 2019 (COVID-19) pandemic has dramatically altered global social environments. The World Health Organization (WHO) declared a global COVID-19 pandemic in March 2020. Subsequently, various restrictions have been recommended worldwide to prevent infection and transmission (e.g., wearing face

**Data availability statement:** All relevant data are within the paper and its Supporting Information files.

**Funding:** This research was supported by a Grant-in-Aid for Scientific Research on Innovative Areas, 'Construction of the Face-Body Studies in Transcultural Conditions' from the Ministry of Education, Culture, Sports, Sciences and Technology (MEXT) KAKENHI (17H06343 to MKY), a Grant-in-Aid for JSPS Transformative Area (A), 'Face-body design: deepening and sublimating face-body based on practical, empirical and constructive research' (25H01235 to MKY), and a Grant-in-Aid for Scientific Research from JSPS (19K14492 and 23H01055 to MK, and 24H00178 to MKY). The funders had no role in study design, data collection and analysis, decision to publish, or preparation of the manuscript.

**Competing interests:** The authors have declared that no competing interests exist.

masks, limiting social interactions, maintaining social distancing, practicing hand hygiene, and imposing travel restrictions). The effects of these changes on mental health and cognitive development have been observed across age groups, including children, adults, and older adults (e.g., [1,2]).

The impact of exposure to faces occluded by masks on the early development of face recognition has raised concerns [3]. Faces are among the most crucial visual stimuli in early life, and convey critical information about other people for the development of social abilities. The most frequent visual stimuli are human faces (e.g., [4,5]), and social cognition relies heavily on facial information [3]. Additionally, previous studies have shown that face processing undergoes significant development during the first year of life and is influenced by early social experiences (e.g., [6,7]). For example, by 3 months of age, infants show a preference for faces that match the gender of their primary caregivers [8]. Around 6 months of age, infants' face processing becomes tuned to faces representative of their social environment, such as own-race faces (e.g., [9]; for review, see [10]).

Moreover, early visual experience has a significant long-term impact on face processing. For example, individuals born with bilateral congenital cataracts who lacked patterned visual input during the early months of life were found to exhibit lasting impairments in second-order relational [11] and holistic processing [12], despite more than 9 years of recovery. Considering these findings, it is crucial to examine the development of face processing in infants predominantly exposed to faces obscured by masks.

A few developmental studies have investigated the effects of face masks on infants' face processing [13,14]. For example, Galusca et al. [14] tested the visual preferences of infants aged 3, 6, 9, and 12 months born during the COVID-19 pandemic. They found that 3- and 6-month-olds looked at upright and inverted masked faces for the same length of time, whereas 9- and 12-month-olds looked for longer at inverted masked faces. The authors suggested that older infants with greater experience with masked faces in general might have become more familiar with upright masked faces. However, whether and how accumulated exposure to masked faces influences infants' face processing remains unclear. Thus, the present study examined face processing in infants born during the COVID-19 pandemic by evaluating their visual preferences for their mothers' faces.

Wearing masks may have significantly affected the development of face perception in infants, particularly in Japan. While mask-wearing in public spaces became common in many countries during the pandemic, its duration and intensity were arguably greater in Japan because of social norms and voluntary adherence, even after official mandates were relaxed (e.g., [15–17]). In Japan, people continued to wear face masks in public for approximately three years following the onset of the pandemic. Consequently, infants born during this period were frequently exposed to masked faces, whereas their exposure to unmasked faces may have been largely limited to their mothers and other household members. These reductions or biases in face exposure have potentially influenced the developmental trajectory of face perception.

In this study, we assessed the visual preferences of 5–8-month-old infants by simultaneously presenting images of their mother and an unfamiliar woman, either with or without face masks. The aim was to examine whether differences in infants' face experiences stemming from regional variations in COVID-19-related restrictions could influence their face preferences. Experiment 1 was conducted in Tokyo, whereas Experiment 2 was conducted in provincial regions (Aichi and Niigata Prefectures).

In Japan, the length, timing, and severity of COVID-19 restrictions, such as stay-at-home requests and inter-prefecture travel limitations, varied by prefecture. When Experiment 1 was conducted, Tokyo was under a state of emergency or quasi-state of emergency, with the government requesting that people voluntarily stay at home. In contrast, during Experiment 2, the provincial regions were not under such declarations, except briefly between January and March 2022. Therefore, families in these regions may experience fewer restrictions on outings and travel than those in Tokyo. Consequently, due to increased social interactions, such as visits to one another's homes, infants in these areas may have been exposed to a greater number of unmasked faces, including those of individuals outside their immediate household.

Previous studies conducted before the COVID-19 pandemic have consistently demonstrated that infants, including newborns [18–20] and those over 6 months [21–23], show a significant preference for their mother's face over unfamiliar faces. Furthermore, infants continued to accumulate visual experiences with their mothers' unmasked faces at home, even during the pandemic. Therefore, we hypothesized that infants in both regions would show a significant preference for their mothers' unmasked faces. However, if regional differences in pandemic-related restrictions influenced infants' face experiences, then their preferences for their mothers' faces might also differ between regions.

Furthermore, Japanese infants born during the pandemic were regularly exposed to their mothers' faces while wearing masks in daily life. Therefore, we also hypothesized that these infants would exhibit a visual preference for their mothers' faces, even when wearing a mask.

Although both national and local governments in Japan requested that people limit their outings during the pandemic, no strict lockdown comparable to those in other countries was imposed. Thus, Japanese families, including those with infants, were still able to leave their homes for essential activities (e.g., shopping, medical visits, or going to parks) during which face masks were worn widely. Accordingly, once outside the home, infants were likely to be exposed to many people wearing masks, and this experience was likely similar across Tokyo and the provincial regions.

## Experiment 1: Testing visual preferences of infants in Tokyo

In Experiment 1, we examined the effects of face masks on infants' visual preferences for their mothers' faces over unfamiliar female faces by simultaneously presenting two faces in the unmasked and masked conditions. As a control, we included an additional object-occluded condition in which faces were occluded by a notebook [24] to determine whether any observed preferences were specifically related to face masks as opposed to facial occlusion in general. Miyazaki and Kawahara [24] reported that adult observers rated faces as less attractive when occluded by a face mask, but not when occluded by a control object such as a notebook, indicating that the occlusion by face masks—which was familiar to Japanese people even before the COVID-19 pandemic—specifically alters face perception. Given this specificity, we hypothesized that infants' responses to their mother's mask-occluded faces would differ from those to object-occluded faces. That is, we expected infants to show a preference for their mothers' faces in the unmasked and masked conditions but not in the object-occluded condition.

## Materials and methods

**Ethics statement.** We explained the study to the infants' parents and obtained their written informed consent. This study was conducted in accordance with the guidelines of the Declaration of Helsinki and approved by the Chuo University Research Ethics Committee (2020−39). The individual in this manuscript has given written informed consent (as outlined in PLOS consent form) to publish case details.

**Participants.** Twenty-two Japanese 5–8-month-old infants born in the West Tokyo District (Tokyo had the highest population density in Japan in 2018, at 6300 per km$^2$ [25]) participated between June and November 2021 (8 males; mean age = 210.21 days; standard deviation (SD) = 33.70; age range 138–251 days). All the participants were healthy full-term infants at the time of testing. Eight other infants were tested but excluded from the final sample because of side bias greater than 90% (i.e., looking at one side of the monitor more than 90% of the time) in either of the three conditions (n = 1), short looking time in either of the three conditions (less than 15 seconds of total looking time; n = 4), or an experimental error (n = 3).

Sample size estimation was conducted using a power analysis based on a previous study [23]. Considering our aim to determine whether infants looked significantly longer at their mother's face than at an unfamiliar female face, we calculated the required sample size for a one-sample $t$-test (i.e., comparing the mean preference score for the mother's face to the 50% chance level). Yamashita et al. [23] reported that effect sizes for infants' preferential looking toward their mother's face over an unfamiliar female face were notably large in 6–7-month-olds (mean Cohen's $d \approx 1.80$). Therefore, we applied a 50% reduced effect (Cohen's $d \approx 0.9$) in our power analysis. This analysis indicated that a minimum sample size of N = 12 would be sufficient to achieve 80% power (1-β = 0.80) at an α level of 0.05 (G*Power; [26]). However, previous studies have recommended a minimum of 20 participants per condition to mitigate the risk of type I errors ( [27]; also see [28] for considerations specific to infant research). Accordingly, we set the sample size to at least 20 participants. We additionally conducted a post hoc power analysis for the within-participant one-way ANOVA (three conditions) using G*Power [26]. Given an alpha level of 0.05, an effect size of f = 0.50 (corresponding to $\eta_p^2 = 0.201$), and a total sample size of 22, the achieved power was 1.00, indicating sufficient sensitivity to detect a large effect.

Participants were recruited through advertisements posted in local information magazines. Only infants whose parents were interested in participating in the study and contacted the laboratory were included. Although we did not collect information on the families' socioeconomic status, the participants would likely have come from a middle-class socioeconomic status. However, given the persistently high rates of face mask use in public areas in Japan (> 90%; [29]) and previous evidence showing no effect of socioeconomic status on infants' visual attention to faces [30], socioeconomic factors are unlikely to have affected the results. Participants were compensated with a gift certificate valued at JP¥2,000 for their involvement in the study.

**Stimuli.** We used full-color images of the frontal view of each infant's mother's face, showing a neutral expression. The images were photographed using a color digital camera (Panasonic DMC-TX1) against an identical white background under identical lighting conditions. When photographing images in the masked condition, the infants' mothers wore identical white face masks. The images for the object-occluded condition were created by overlaying the image of a brown notebook (visual angle: approximately 13.5° in width and 9.8° in height) on each unmasked face image to obscure the same area of the face as in the masked face images (for details, see [24]). Considering previous findings indicating that infants do not show a color preference for brown [31], the color may have little to no effect on the results. The three types of face images were cropped along hairlines and facial contours using the Gnu Image Manipulation Program (GIMP) (Fig 1). For the unfamiliar face image, the image of a different mother whose hairstyle was similar to that of the infant's mother was randomly chosen for each infant. Although we did not systematically control for facial similarities between the mother and an unfamiliar female, all face stimulus pairs fell within the range of natural variation.

Each facial image was subtended by a visual angle of approximately 12.34° in width and 17.4° in height when the infants viewed at the images from approximately 40 cm away. The image of the infant's mother's face and that of an unfamiliar female face were approximately 16.1° apart when placed side by side on the monitor.

**Apparatus.** Each infant was seated on their parent's lap in a booth during the experiment. The booth was covered with a black cloth to minimize distraction from the monitor. Visual stimuli were presented on a 23.8-inch LCD monitor (EIZO FlexScan EV2451, Japan) set at a resolution of 1920 (horizontal) × 1080 (vertical) pixels in 8-bit color mode using PsychoPy v 1.90.1 [32]. Two loudspeakers were placed at the sides of the monitor to present sounds. The infants' viewing

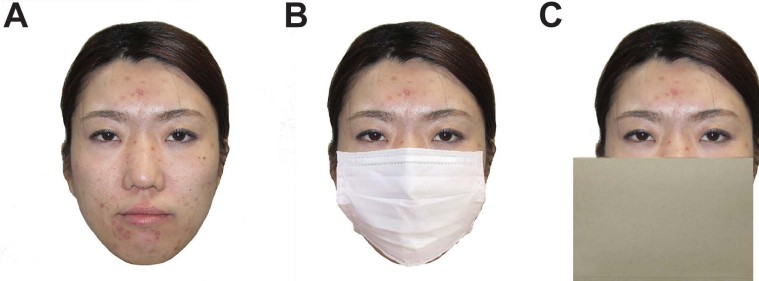

**Fig 1. Examples of face stimuli used in the unmasked (a), masked (b), and object-occluded (c) conditions.** Informed consent for the publication of identifying facial images in an online open-access publication was obtained.

behavior during the experiment was monitored and recorded using a camera (Logicool C920R) located under the center of the monitor. The experimenter controlled the timing of the presentation of the fixation and facial stimuli, guided by a live image of the infant's behavior captured by the camera.

During the experiment, the parents were instructed to close their eyes to avoid looking at the monitor or talking to their infants. The parents were blinded to the hypothesis of the study and the predicted looking behavior of the infant in each experimental condition.

**Procedure.** Each infant was tested using a preferential-looking procedure comprising two 15-second trials in each of the three test conditions (90 seconds in total). Each trial began with a cartoon image as a fixation point appearing at the center of the monitor, accompanied by a beep to attract the infants' attention. As soon as the infants started viewing the fixation image, the experimenter replaced it with the face stimuli for 15 seconds. The duration of stimulus presentation was fixed at 15 seconds, regardless of whether the infants looked. For each trial in each condition, images of the infant's mother's face and an unfamiliar female's face were presented side by side on the monitor. The positions of the two images were counterbalanced across infants in the first trial and reversed in the second trial. The presentation order of the three conditions (unmasked, masked, and object-occluded) was also counterbalanced across participants.

**Analysis.** An observer analyzed the video recordings of the infants' viewing behavior frame-by-frame and recorded the looking time for each of the two images in every trial. A second independent observer analyzed 40% of the total data for unmasked, masked, and object-occluded conditions. Neither observer knew the face image conditions or the position of the target image (left or right) on the monitor. The inter-observer reliability on the total looking times for the mother's and an unfamiliar female faces was excellent throughout the experiment (Pearson's $r=0.97$).

We defined the image of the infant's mother's face as the target. We calculated the individual percentage preference scores by dividing each infant's looking time at their mother's face during the test trial by the total looking time over the test trial and then multiplying the ratio by 100. We determined whether infants showed a significant preference for their mother's faces by conducting two-tailed one-sample $t$-tests against a 50% the chance level with a Bonferroni correction ($\alpha$ level of 0.05/3 = 0.017) for multiple comparisons of the mean preference scores for the mother's faces in the three conditions. We applied the Bonferroni correction rather than alternative methods (e.g., Holm and False Discovery Rate) for two main reasons. First, the sample sizes in infant perception research tend to be small because of practical constraints, although the current sample sizes were determined based on power analyses. Small samples increase the risk of spurious findings owing to random variation; however, the Bonferroni correction is highly conservative and helps reduce the likelihood of Type I errors (false positives). Second, this study involved a small predefined set of comparisons. Under such conditions, the increase in Type II errors (false negatives) associated with the Bonferroni correction is minimal compared to other correction methods. Moreover, the Bonferroni correction is widely used in developmental research on infant perception.

## Results and discussion

Fig 2A shows the mean preference scores for their mothers' faces of infants raised in Tokyo. Two-tailed one-sample *t*-tests versus chance level (50%) revealed that infants looked significantly longer at their mother's faces in the masked condition ($M = 58.29\%$, $t(21) = 4.66$, $p < 0.01$, Cohen's $d = 1.40$). In contrast, no statistically significant preference was observed in the unmasked condition ($M = 50.46\%$, $t(21) = 0.21$, $p = 0.83$, Cohen's $d = 0.07$). In the object-occluded condition, infants' looking times did not differ significantly between their mother's and unfamiliar female faces ($M = 51.14\%$, $t(21) = 0.63$,

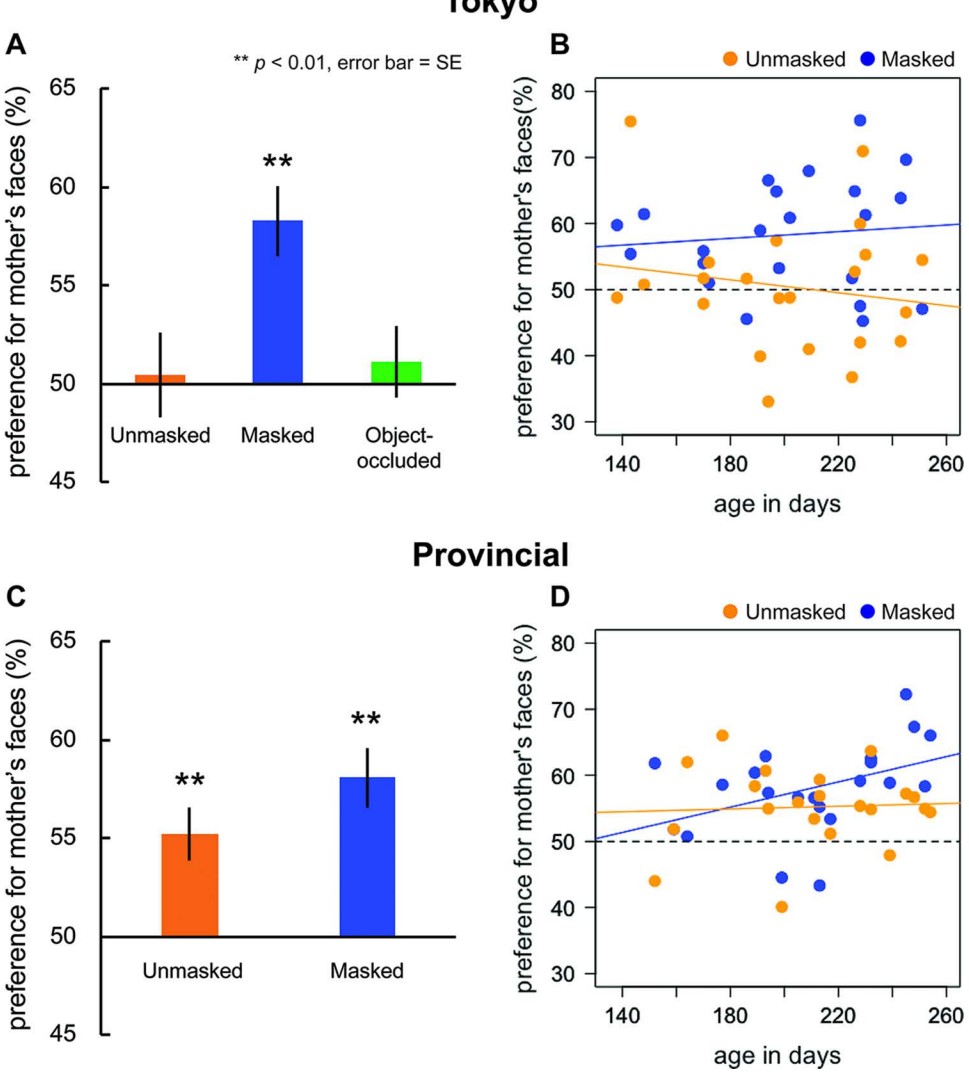

**Fig 2. Infants' preference for their mother's faces in Tokyo and provincial cities. (A & C)** Mean preference scores for the mother's faces in three conditions (unmasked, masked, and object-occluded) in Tokyo (2A) and in two conditions (unmasked and masked) in provincial cities (2C). The mean preference scores are shown in orange bars for the Unmasked condition, in blue bars for the Masked condition, and in green bars for the Object-occluded condition. The error bars indicate mean standard errors (SE). Significant differences versus chance level (50%) are marked with asterisks (*p*-value for multiple comparisons are adjusted using a Bonferroni correction, α level of 0.05/the number of conditions). **(B & D)** Individual data showing preference for the mother's face are plotted as a function of the infant's age. The orange and the blue circles represent preference scores in the unmasked and masked conditions, respectively. The orange line is the regression line fitted to the orange circles (Tokyo: $R^2 = 0.03$, Provincial: $R^2 = 0.03$), and the blue line is the regression line fitted to the blue circles (Tokyo: $R^2 = 0.01$, Provincial: $R^2 = 0.18$).

$p = 0.53$, Cohen's $d = 0.19$). A one-way ANOVA with conditions (masked, unmasked, and object-occluded) revealed a significant main effect of condition ($F(2, 42) = 5.29$, $p < 0.01$, $\eta^2 = 0.14$), with significantly higher preference score in the masked condition than those in the unmasked ($p < 0.05$, Cohen's $d = 0.87$) and object-occluded conditions ($p < 0.05$, Cohen's $d = 0.80$), respectively.

Fig 2B presents individual data on each infant's preference scores for the mothers' faces as a function of age. We calculated the correlation between preference scores and age in days and tested the correlation coefficient in both the masked and unmasked conditions to examine the trend in infants' preference scores. Although it appears that preference scores in the unmasked condition decreased with increasing age, we found no significant correlations between age and preference scores in either the unmasked ($r = -0.16$, $p = 0.47$) or masked ($r = 0.10$, $p = 0.65$) conditions.

We found that 5–8-month-old infants born and raised in Tokyo during the COVID-19 pandemic showed a preference for their mothers' faces if occluded by a mask, but not by objects. These results suggest that infants' visual preferences do not result solely from face occlusion. Moreover, the finding that infants preferred the masked faces of their mothers suggests that they adapted to the social environment specific to the COVID-19 pandemic, in which people around them, including their mothers, wore face masks outside the household. This is consistent with previous findings that even at 3 months of age, face processing ability develops to adapt to the social environments around individual infants by showing preferences for face categories with which they have more visual experiences (i.e., the gender of the primary caregiver [8] and own-race faces [33–35]).

The finding that infants in Tokyo did not prefer their mothers' unmasked faces is inconsistent with previous studies that have consistently reported a preference for mothers' faces among infants [21–23]. Furthermore, this contradicts a recent study conducted in another prefecture of Japan, which found that infants significantly preferred their mothers' unmasked faces during a period similar to that of the COVID-19 pandemic [36]. The methodologies of these studies were largely comparable, including infant age, stimulus size, and duration of stimulus presentation. The only notable difference was in the color of face stimuli. This study used full-color images, whereas some previous studies [21,22] used grayscale images. However, the use of full-color images is unlikely to have led to the null result, as full-color images provide richer visual information than grayscale images.

A closer examination of the individual data (Fig 2B) suggested that the infants in Tokyo were not necessarily incapable of differentiating between their mothers and unfamiliar female faces. Individual preference scores were not clustered around 50% but instead varied widely across participants. One possible explanation is that differences in infants' face exposure due to COVID-19 restrictions may have influenced their face preferences.

In Experiment 2, we explored this possibility by examining infants born and raised in regions outside Tokyo. As noted in the Introduction, the length, timing, and severity of the COVID-19 restrictions (e.g., the state of emergency) varied across prefectures. In provincial areas, where fewer cases were reported, restrictions were typically shorter and less stringent than those in Tokyo. Furthermore, extended-family households are more common in the provincial regions than in Tokyo, where nuclear families are more prevalent. Living near relatives is not unusual among families raising infants in these regions, potentially increasing opportunities for social interaction without face masks.

Considering these factors, infants in provincial areas may have accumulated different face experiences than those in Tokyo. Therefore, we hypothesized that their preference patterns for masked and unmasked faces would also differ. Specifically, based on previous findings showing that infants in non-Tokyo prefectures preferred their mothers' unmasked faces, even during the pandemic [36], we expected them to show a preference for their mothers' faces in both the unmasked and masked conditions.

Experiment 2 did not include the object-occluded condition because Experiment 1 demonstrated that infants' visual preference for their mothers' masked faces was not merely due to partial occlusion. During the COVID-19 pandemic, mothers' masked and unmasked faces were common in everyday life in both Tokyo and provincial areas, and thus were likely familiar to infants. By contrast, the notebook-occluded condition used in Experiment 1 was intended to present an

unfamiliar visual stimulus. Unlike face masks, a notebook covering the face does not resemble typical social visual input, and is therefore expected to be equally unfamiliar to infants across regions.

### Experiment 2: Testing visual preferences of infants in the provincial areas of Japan

**Materials and methods.** The methods used for the experiments with infants in the provincial cities (Kasugai City in Aichi Prefecture and Niigata City in Niigata Prefecture) were the same as those used for infants in Tokyo, except for the following reasons: Aichi and Niigata had much lower population densities than Tokyo in 2018 (1457 per $km^2$ for Aichi and 179 per $km^2$ for Niigata, respectively [25]).

**Ethics statement.** This study was conducted in accordance with the guidelines of the Declaration of Helsinki and approved by the Ethical Committee of the Aichi Developmental Disability Center (1−21) and the Ethical Committee of Niigata University (2022−0003). Written informed consent was obtained from the parents of all infants.

**Participants.** A total of twenty-one Japanese 5–8-month-old infants participated between January and March 2022 in Aichi Prefecture (9 infants from Kasugai City) and between December 2022 and February 2023 in Niigata Prefecture (12 infants from Niigata City; 12 male; mean age = 210.29 days; SD = 29.98; age range 152–254 days). When Experiment 2 was conducted, Aichi and Niigata Prefectures were not in a state of emergency or quasi-state of emergency, except between January and March 2022. Therefore, people in these two prefectures were not restricted from social interactions, outings, or travel across the prefectural borders. However, they were still asked to continue wearing face masks outside their homes. No infants were excluded from the final sample.

Participants were recruited through advertisements posted in local information magazines or leaflets distributed via public health service centers. It is assumed that, based on the voluntary nature of participation, most participants came from families with middle-class socioeconomic backgrounds in Japan.

**Stimuli.** We used full-color images of the frontal view of each infant's mother's face showing a neutral expression. The images used in the masked and unmasked conditions were photographed using a digital camera (Canon IXY 400F). These images were cropped along the hairlines and facial contours using Adobe Photoshop.

**Apparatus.** Visual stimuli were presented on a 23.8-inch LCD monitor (BenQ GW2470) set at a resolution of 1920 (horizontal) × 1080 (vertical) pixels. The infants' viewing behavior during the experiment was monitored and recorded using a charge-coupled device camera located under the center of the monitor.

**Procedure.** Similar to Experiment 1, each infant was tested using a preferential-looking procedure comprising of two 15-second trials; however, it included only the masked and unmasked conditions (60 seconds in total).

**Analysis.** An observer analyzed the video recordings of the infants' viewing behavior frame-by-frame and recorded the looking time for each of the two images in each trial. A second independent observer analyzed 40% of the total data for both the unmasked and masked conditions. The inter-observer reliability of the total looking time infants looked at their mothers and unfamiliar female faces throughout the experiment was excellent (Pearson's $r = 0.90$). Two-tailed one-sample $t$-tests against the 50% chance level were conducted to determine whether infants showed a significant preference for their mother's faces, with the α level Bonferroni-adjusted to 0.05/2 = 0.025.

### Results and discussion

[Fig 2C] presents the mean preference scores for their mothers' faces of infants in provincial cities. Two-tailed one-sample $t$-tests versus chance level (50%) on the mean preference scores for their mother's faces revealed that infants showed a significant preference for their mother's faces in the masked ($M = 58.08\%$, $t(20) = 5.35$, $p < 0.01$, Cohen's $d = 1.65$) and unmasked ($M = 55.22\%$, $t(20) = 3.93$, $p < 0.01$, Cohen's $d = 1.21$) conditions. We found no significant difference in face preference between the two test conditions ($t(20) = 1.65$, $p = 0.11$, Cohen's $d = 0.45$, two-tailed).

Individual data showing mothers' face preferences for infants born and raised in the provincial cities of Japan were plotted as a function of age in days ([Fig 2D]). The preference scores appeared to increase with age in the masked condition

but not in the unmasked condition. We calculated the correlations between preference scores and age in days, and tested the correlation coefficient to investigate whether any developmental changes could be observed. No significant correlations were found for the unmasked ($r = 0.05$, $p = 0.82$) and masked ($r = 0.42$, $p = 0.05$) conditions.

As shown in Figs 2B and 2D, the individual preference scores of infants in Tokyo appear to be more highly scattered than those of infants in provincial cities. We tested this by applying $f$-tests to compare the variance in the mean preference scores between both areas of Japan in the masked and unmasked conditions. We found that in the unmasked condition, individual preference scores were significantly more scattered in Tokyo than in the provincial cities (Variance $_{Tokyo}$ = 100.25, Variance $_{Provincial}$ = 37.02, $p = 0.02$). However, the $f$-test revealed that the variances were not significantly different between Tokyo and the provincial cities in the masked condition (Variance $_{Tokyo}$ = 69.70, Variance $_{Provincial}$ = 47.91, $p = 0.20$).

In Experiment 2, we found that, unlike infants in Tokyo, those in provincial cities preferred their mothers' faces to an unfamiliar female face, regardless of whether the faces were masked. Moreover, infants in provincial cities showed a more consistent preference for their mothers' unmasked faces than did infants in Tokyo. Although the infants who participated in Experiments 1 (Tokyo) and 2 (provincial cities) were all born and raised during the pandemic, as predicted, their preference patterns for their mothers' faces differed. This difference may be because infants in non-Tokyo provincial areas were able to accumulate more experiences of faces not wearing face masks than Tokyo infants due to the shorter periods of COVID-19 restrictions.

However, neither Experiment 1 nor 2 examined the experiences of individual infants with faces. We further explored the relationship between face experiences and mothers' face preferences by surveying infants who participated in Experiments 1 and 2. We focused on infants' individual experience of seeing unmasked faces of people other than their mothers because we predicted that experiences with unmasked faces would vary more across infants than experiences with masked faces. When Experiments 1 and 2 were conducted, almost all individuals in Japan, including mothers of infants, wore face masks outside their homes. Therefore, the infant participants in Tokyo and the provincial cities had similar experiences with masked faces, seeing their mothers and other people wearing masks outside their homes. In contrast, exposure to unmasked faces would vary across infants, depending on the number of family members living together and interactions with non-family members who visited the infants' homes without wearing face masks.

## Experiment 3: Survey of infants' facial experiences

We conducted a survey to ask parents about their infants' experiences of seeing unmasked faces from birth to the day of the experiment to further understand the observed differences in preferences for their mothers' unmasked faces. We examined the relationship between infants' individual face experiences and their preferences for their mothers' unmasked faces, as shown in Experiments 1 and 2.

For most of the period during which Experiment 1 was conducted (June–November 2021), Tokyo was in a state of emergency or quasi-state of emergency. In contrast, during most of the periods when Experiment 2 was conducted (January–March 2022 and December 2022–February 2023), the provincial regions were not under such declarations. Considering these situational differences, it is possible that the families of infant participants in Tokyo were more likely to refrain from going out than those in the provincial regions.

However, during both experimental periods, although the Japanese or local governments requested people to limit their outings, no strict lockdown, as seen in other countries, was in place. Thus, Japanese people, including families with infants, were still able to go out for essential activities (e.g., shopping, medical visits, or going to parks). During these outings, people generally wore face masks. Therefore, once outside the home, it is reasonable to assume that infants were exposed to many masked faces and that this experience was generally similar across Tokyo and the provincial regions.

## Materials and methods

**Ethics statement.** The current experiment was conducted in accordance with the Declaration of Helsinki and was approved by the Ethical Committee of Niigata University (2022−0003). All the parents of infants reviewed the written electronic consent form online and provided electronic consent before beginning the procedure.

**Participants.** We sent online questionnaires to the families who participated in Experiments 1 (Tokyo) and 2 (provincial cities) in February 2023. Six families in Tokyo and 16 families in provincial cities (Aichi and Niigata) responded.

The questionnaire did not collect any demographic data, including the respondents' identities (mother or father). However, it can reasonably be assumed that most respondents to the questionnaire in Experiment 3 were mothers for several reasons. In Japan, most working mothers (approximately 80–85%) take parental leave, with approximately 90% of them doing so for more than 6 months. In contrast, only approximately 10% of fathers take parental leave, although this number has been gradually increasing in recent years. Furthermore, the duration of leave taken by fathers tends to be significantly shorter than that of mothers, with approximately 90% of fathers taking less than 3 months of leave [37]. Consequently, infants under 1 year of age in Japan spend most of their early lives with their mothers, regardless of whether they are employed.

**Materials.** The infants' parents were asked to complete an online questionnaire reporting their infants' experience with people not wearing face masks from birth to participation in the experiment. Specifically, the parents responded to questions about the number of family members living together, the number of relatives (e.g., grandparents, etc.) and non-relatives (e.g., parents' friends) who visited their households and interacted with their infants, and the number of people with whom the infants interacted when visiting relatives' and non-relatives' homes. We also asked how often the people who interacted with their infants (family members, relatives, and non-relatives) wore face masks. The parents indicated the frequency as "every time," "almost every time," "neither," "almost never," or "never." Further details regarding these questions can be found in Supporting Materials (S3 File).

## Results and discussion

If infants' parents indicated that a person/people "never" or "almost never" wore a face mask, they were considered as a person/people who did not wear a face mask. We then calculated the number of unmasked individuals to whom the infants were exposed. This included family members living together, relatives, and non-relatives. Table 1 shows the number of unmasked faces the infants saw from birth to the time of participation in the experiment, along with their preference scores for the mother's face. Although we obtained a limited number of responses from parents, it appeared that if infants saw a smaller number of unmasked faces, especially fewer than nine people, they would prefer an unfamiliar female face to their mother's face. In contrast, if the infants saw a greater number of unmasked faces, they consistently demonstrated a preference for their mothers. This was observed regardless of the infant's age or whether they were born and raised in Tokyo or provincial cities.

We used an independent-sample $t$-test to compare the mean number of unmasked faces that infants viewed in Tokyo and the provincial cities. The results showed no significant differences between the groups ($M_{Tokyo} = 20.67$, $M_{Provincial} = 22.94$, $t(6) = 0.16$, $p = 0.88$, Cohen's $d = 0.09$). If we divided the infants into two groups, the mother-preferred group (showing a mother's face preference score over 50% in the unmasked condition, n = 16) and the unfamiliar-preferred group (showing a mother's face preference below 50%, n = 6), the mother-preferred group accumulated more experience with unmasked individuals ($M = 28.63$) than the unfamiliar-preferred group ($M = 5.5$, *independent-sample t-test:* $t(18) = 3.73$, $p = 0.002$, Cohen's $d = 1.13$). However, we found no significant between-group differences in the number of family members living together ($M_{mother-prefer group} = 3.31$, $M_{mother-prefer group} = 3.0$, *independent-sample t-test: $t(16) = 0.59$, $p = 0.56$, Cohen's $d = 0.22$*). Furthermore, the parents in either group did not report that family members living together wore face masks when they interacted with their infants at home. Collectively, these comparisons showed that the experience of seeing unmasked faces, especially those of people who did not live together, might influence whether infants preferred their mothers or unfamiliar faces.

**Table 1. The number of unmasked faces that infants were exposed to and preference scores for the mother's face in the Unmasked condition for each participant.**

| ID | Age (days) | Region | Number of seeing unmasked faces | Number of family member | Preference scores in the Unmasked condition (%) |
|---|---|---|---|---|---|
| 1 | 239 | P | 2 | 2 | 47.89 |
| 2 | 245 | T | 2 | 2 | 46.55 |
| 3 | 243 | T | 3 | 3 | 42.18 |
| 4 | 213 | P | 4 | 2 | 59.32 |
| 5 | 225 | T | 5 | 3 | 36.75 |
| 6 | 191 | T | 6 | 4 | 39.91 |
| 7 | 205 | P | 9 | 4 | 55.91 |
| 8 | 232 | P | 10 | 4 | 63.67 |
| 9 | 159 | P | 11 | 3 | 51.85 |
| 10 | 189 | P | 12 | 2 | 58.36 |
| 11 | 199 | P | 15 | 4 | 44.52 |
| 12 | 213 | P | 17 | 2 | 56.87 |
| 13 | 211 | P | 18 | 8 | 53.41 |
| 14 | 248 | P | 18 | 3 | 56.69 |
| 15 | 194 | P | 20 | 3 | 54.95 |
| 16 | 232 | P | 20 | 2 | 54.85 |
| 17 | 172 | T | 24 | 2 | 54.11 |
| 18 | 177 | P | 40 | 4 | 66.02 |
| 19 | 228 | P | 47 | 3 | 55.36 |
| 20 | 217 | P | 59 | 3 | 51.16 |
| 21 | 245 | P | 65 | 3 | 57.22 |
| 22 | 230 | T | 84 | 5 | 55.27 |

Note. T indicates infants who were raised in Tokyo (participants in Experiment 1), and P indicates infants who were raised in the provincial cities (participants in Experiment 2).

## General Discussion

This study examined face processing in infants born and raised during the COVID-19 pandemic by evaluating their visual preferences for their mothers' faces compared with unfamiliar faces, both with and without face masks. We found that infants in Tokyo and those in provincial cities showed a significant preference for their mothers' faces when presented with images of masked faces from as early as 5 months of age. This finding suggests that Japanese infants born during the pandemic were able to adapt to a social environment in which faces around them were frequently covered with masks.

Previous studies have shown that infants' face processing is shaped by the dominant face categories in their social environments [8,33–35]. For example, infants as young as 3 months of age exhibited a preference for faces matching the gender of their primary caregiver [8] and faces of their own race [33–35]. At approximately 6 months of age, infants begin to show better recognition of own-race faces than other-race faces, a phenomenon known as perceptual narrowing (for a review, see [10]).

These findings were consistent with our results. This study further supports the developmental perspective in which face processing is shaped by visual experiences, particularly the so-called "face-diet," which refers to the collection of faces infants encounter in everyday life (e.g., [38,39]).

Additionally, we found that infants in provincial cities (Aichi and Niigata Prefectures) showed a clear visual preference for their mothers' faces in the unmasked condition, whereas infants in Tokyo did not. One potential explanation for this discrepancy is the difference in the COVID-19-related restrictions between the two settings. Although face masks were

required in public across all sites, the timing and severity of restrictions, such as stay-at-home orders and inter-prefecture travel limitations, were more stringent and prolonged in Tokyo than in Aichi and Niigata Prefectures. Consequently, infants in provincial regions may have had more opportunities to see unmasked faces at home, particularly through interhousehold interactions with extended family members and friends. This interpretation is supported by our parental questionnaire data, which suggests that greater exposure to unmasked faces is associated with a preference for the mother's face.

One possible explanation for why some infants with limited exposure to unmasked faces did not show a preference for their mothers' faces is that they may have become overly familiar with them. A previous study demonstrated that infants' initial preferences for their mothers' faces shifted to a novel, unfamiliar face after repeated exposure to their mother's face [40]. Generally, infants' looking preferences often shift from familiarity to novelty as exposure increases, regardless of stimulus type [41–43]. Thus, it is plausible that the primary unmasked face these infants saw was that of their mothers, and extensive exposure may have led to a novelty preference in the unmasked condition. However, the current study did not collect detailed data on the amount of time infants spent with their mothers and others without face masks or on the quality of those interactions. Future research could use methods such as head-mounted cameras to record infants' face-to-face experiences to test this hypothesis more directly.

In contrast, infants who showed a preference for their mothers' faces in the unmasked condition had been exposed to a greater number of unmasked individuals, on average, approximately 30 different faces without masks, albeit with variability (Table 1). This level of face exposure was comparable to that in infants born before the pandemic. For example, a study using head-mounted cameras found that infants born before the pandemic viewed an average of 51 individual faces over 10 days [44]. Moreover, those infants spent approximately 57.53% of their face-viewing time with their mothers and 40% with others (e.g., non-primary caregivers, relatives, and strangers) in their homes. Although differences in methodology and timeframes make direct comparisons difficult, infants who preferred their mothers' unmasked faces in our study may have experienced similar diversity and richness in exposure to unmasked faces.

## Limitations and implications

Notably, the questionnaire used in this study was administered retrospectively and not on the same day as the visual preference test. Furthermore, the number of responses was limited, particularly from parents in Tokyo. Therefore, we caution against drawing strong conclusions regarding the relationship between infants' face exposure and visual preferences based solely on these data. Smaller sample sizes reduce the statistical power and may introduce bias. However, the data in Table 1 suggest that, even in Tokyo, infants with greater exposure to unmasked faces were more likely to show a preference for their mothers' faces. Conversely, in provincial regions, infants with limited exposure to unmasked faces did not show such a preference despite looser COVID-19 restrictions. These findings highlight the importance of investigating individual differences in visual experience using a larger sample.

Future studies should also consider the potential influence of socioeconomic factors, as they may be related not only to mask-wearing behavior but also to infants' face processing itself. For example, socioeconomic factors could influence the frequency and quality of social interactions. Although most participants in the present study were likely from middle-class families due to the voluntary nature of participation, examining infants from a wider range of socioeconomic backgrounds would help clarify how mask use may affect the development of face perception.

In this study, the amount and quality of maternal interaction without a mask may have been relatively consistent across regions because mothers are typically the primary caregivers in Japanese households. However, considering previous findings that infants' face preferences can be influenced by caregiver characteristics (e.g., sex of the primary caregiver; [8]), our findings should be generalized to more diverse caregiving contexts with caution.

Another limitation of this study is that interobserver reliability was calculated for 40% of the dataset rather than the entire sample. Although this approach is consistent with previous infant research practices, it may limit the generalizability of the reliability estimates.

 

## Conclusion

In conclusion, this study used a preferential-looking method to investigate 5–8-month-old infants' preferences for their mothers' faces during the COVID-19 pandemic. In contrast to pre-pandemic findings that consistently showed a preference for the mother's face (e.g., [21]), infants' preferences in our study varied. This variability may be related to differences in COVID-19 restrictions and/or exposure to unmasked faces from birth. These findings suggest that individual differences in face experience in everyday social environments may modulate the development of face processing in infancy.

## Supporting information

**S1 Data. Raw data of infants' looking time for Experiment 1.**
(XLSX)

**S2 Data. Raw data of infants' looking time for Experiment 2.**
(XLSX)

**S3 File. The details of the online questionnaire of Experiment 3.**
(DOCX)

## Acknowledgments

The authors thank Y. Nakashima, Y. Tsuji, S. Tsurumi, N. Yamanaka, and J. Yang for their assistance with the data collection. We specifically thank the infants and their parents for their kindness and cooperation during the study.

## Author contributions

**Conceptualization:** Megumi Kobayashi, Machi Sugai, So Kanazawa, Masami K. Yamaguchi.

**Data curation:** Megumi Kobayashi, Machi Sugai.

**Formal analysis:** Megumi Kobayashi, Machi Sugai.

**Funding acquisition:** Megumi Kobayashi, Masami K. Yamaguchi.

**Investigation:** Megumi Kobayashi, Machi Sugai.

**Methodology:** Megumi Kobayashi, Machi Sugai, So Kanazawa.

**Project administration:** Masami K. Yamaguchi.

**Supervision:** So Kanazawa, Masami K. Yamaguchi.

**Validation:** Megumi Kobayashi.

**Visualization:** Megumi Kobayashi.

**Writing – original draft:** Megumi Kobayashi.

**Writing – review & editing:** Megumi Kobayashi, So Kanazawa, Masami K. Yamaguchi.

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
