## [Decision Letter · Decision Letter 0]

7 Apr 2025

Dear Dr. Kobayashi,

Thank you for submitting your manuscript to PLOS ONE. After careful consideration, we feel that it has merit but does not fully meet PLOS ONE’s publication criteria as it currently stands. Therefore, we invite you to submit a revised version of the manuscript that addresses the points raised during the review process.

We look forward to receiving your revised manuscript.

Kind regards,

Tadashi Ito

Academic Editor

PLOS ONE

Journal Requirements:

4. We note that Figure 1 includes an image of a participant  in the study.

5. Please include captions for your Supporting Information files at the end of your manuscript, and update any in-text citations to match accordingly. Please see our Supporting Information guidelines for more information: http://journals.plos.org/plosone/s/supporting-information .

6. We note that there is identifying data in the Supporting Information file SupplementalFile1_Exp1data.xlsx and SupplementalFile2_Exp2data.xlsx. Due to the inclusion of these potentially identifying data, we have removed this file from your file inventory. Prior to sharing human research participant data, authors should consult with an ethics committee to ensure data are shared in accordance with participant consent and all applicable local laws.

-Location data

Additional Editor Comments:

Please make revisions according to the reviewers' suggestions.

Reviewers' comments:

Reviewer's Responses to Questions

**Comments to the Author**

1. Is the manuscript technically sound, and do the data support the conclusions?

Reviewer #1: Partly

Reviewer #2: Yes

2. Has the statistical analysis been performed appropriately and rigorously?

Reviewer #1: Yes

Reviewer #2: No

3. Have the authors made all data underlying the findings in their manuscript fully available?

Reviewer #1: Yes

Reviewer #2: Yes

4. Is the manuscript presented in an intelligible fashion and written in standard English?

Reviewer #1: Yes

Reviewer #2: Yes

Reviewer #1: see Review Document

Reviewer #2: The authors present very interesting findings based on two experimental studies and a follow-up survey on infant’s preferences for mother’s face during COVID-19 pandemic. Below are my suggestions for further improvement of the manuscript before I can endorse publication:

1. You are referring to a variety of facial experiences without necessarily implying a switch or alternation. Perhaps, a word like "variation" might better convey your intended meaning. Please consider changing the beginning of the title as “variation in face experiences ……”

2. Please elaborate on how the sample size was determined for Experiments 1 and 2. Did you perform a power analysis?

3. Lines 218- 225: Please elaborate further on how previous studies differed from the present study, and how these differences might explain the inconsistencies between your findings and those of earlier reports.

4. Lines 164 and 277: Did I understand correctly that the experiments consisted of only 2 trials? Could you please include the rationale for having very few trials?

5. Line 176- 181: It is great that you have ensured interobserver reliability, however, why is it that only 40% of the data was analysed by the second observer, rather than two independent observers analyzing the entire dataset? Ideally, the second observer should have completed the analysis of the remaining 60%. If this is not feasible, the rationale behind this decision must be addressed and referred to as a limitation in the discussion section, as this can potentially influence the results.

6. Lines 193: 196: In this section, there are some issues related to interpretation of the non-significant findings. P-values should not be interpreted to confirm the null, as these are likely inconclusive findings (considering the power& sample size). I suggest either following up these findings with Bayesian statistics or rephrasing the sentences in a way that they do not confirm the null.

Minor:

Line 97: Don’t start the sentence with also

Lines 104- 107: Please rephrase this sentence for an improved grammar and clarity.

Line 153: “brown-colored notebook”, which appears green in the image in Figure 1.

Line 199: Please include cohen’s d here too.

Line 239: Replace the word “facts” with “findings”

**Do you want your identity to be public for this peer review?** For information about this choice, including consent withdrawal, please see our Privacy Policy

Reviewer #1: No

Reviewer #2: No

---

## [Author Response · Author response to Decision Letter 1]

15 Oct 2025

Thank you for your thoughtful and constructive feedback. Please refer to the point-by-point response document for detailed explanations of how each comment was addressed.

---

## [Decision Letter · Decision Letter 1]

4 Nov 2025

Dear Dr. Kobayashi,

Thank you for submitting your manuscript to PLOS ONE. After careful consideration, we feel that it has merit but does not fully meet PLOS ONE’s publication criteria as it currently stands. Therefore, we invite you to submit a revised version of the manuscript that addresses the points raised during the review process.

We look forward to receiving your revised manuscript.

Kind regards,

Tadashi Ito

Academic Editor

PLOS ONE

Journal Requirements:

Reviewers' comments:

Reviewer's Responses to Questions

**Comments to the Author**

Reviewer #1: All comments have been addressed

2. Is the manuscript technically sound, and do the data support the conclusions?

Reviewer #1: Yes

3. Has the statistical analysis been performed appropriately and rigorously?

Reviewer #1: Yes

4. Have the authors made all data underlying the findings in their manuscript fully available?

Reviewer #1: Yes

5. Is the manuscript presented in an intelligible fashion and written in standard English?

Reviewer #1: Yes

Reviewer #1: Thank you very much for sharing the revised version of your manuscript. We can see that you have implemented all of our previous comments, which has improved the overall clarity of the paper. We still have a few remaining points that we would like to raise:

1) Hypothesis (126–131, page 6):

The description of the findings by Miyazaki and Kawahara [24] and the subsequent hypothesis are somewhat confusing. The cited study reports that observers rated faces as less attractive when occluded by a mask, but not when occluded by a control object (e.g., a notebook). Based on this finding, one would expect that infants might prefer their mother’s face occluded by an object rather than by a mask, which seems the opposite of what is stated in your hypothesis. As currently written, it is not entirely clear how your hypothesis logically follows from the cited finding. Perhaps you could clarify this link or adjust the phrasing to better reflect the underlying rationale.

2) Power analysis:

You mention that the power analyses were conducted for t-tests, but the largest statistical model used in the study was an ANOVA. Could you please clarify whether the power was sufficient for this more complex model as well?

3) SES argument (333–334, page 15):

You wrote, “All participants came from families with a middle-class socioeconomic background in Japan.” However, as you mentioned on page 8, this is only an assumption based on the voluntary nature of participation. It might be important to make this explicit. The same applies to lines 551–553: “This is because they could be related to mask-wearing behavior, although the participants of this study had a middle-class socioeconomic background.”

4) SES argument (162-167, page 8):

You discuss socioeconomic status (SES) in relation to mask frequency, but SES could also lead to differences in face perception itself. In other words, the same frequency of exposure to masked faces might have different effects on children from different SES backgrounds. It would be interesting if you could briefly discuss this possibility.

5) Typo (409, page 18): There seems to be a small typo, I assume the year should be 2021 and not 2011.

**Do you want your identity to be public for this peer review?** For information about this choice, including consent withdrawal, please see our Privacy Policy

Reviewer #1: No

---

## [Author Response · Author response to Decision Letter 2]

17 Nov 2025

Thank you for your thoughtful comments. Please see attached file.

---

## [Decision Letter · Decision Letter 2]

23 Dec 2025

Variations in face experiences during the COVID-19 pandemic affect infants’ preference for their mother’s face

PONE-D-24-49778R2

Dear Dr. Megumi Kobayashi,

We’re pleased to inform you that your manuscript has been judged scientifically suitable for publication and will be formally accepted for publication once it meets all outstanding technical requirements.

Kind regards,

Tadashi Ito

Academic Editor

PLOS One

Additional Editor Comments (optional):

Reviewers' comments:

Reviewer's Responses to Questions

**Comments to the Author**

Reviewer #1: All comments have been addressed

2. Is the manuscript technically sound, and do the data support the conclusions?

Reviewer #1: Yes

3. Has the statistical analysis been performed appropriately and rigorously?

Reviewer #1: Yes

4. Have the authors made all data underlying the findings in their manuscript fully available?

Reviewer #1: Yes

5. Is the manuscript presented in an intelligible fashion and written in standard English?

Reviewer #1: Yes

Reviewer #1: Thank you for sharing the revised version of your manuscript. I can confirm that all of our previous comments have been taken into account. The revision has significantly improved the clarity and overall quality of the paper.

**Do you want your identity to be public for this peer review?** For information about this choice, including consent withdrawal, please see our Privacy Policy

Reviewer #1: No

---

## [Editor Report · Acceptance letter]

PONE-D-24-49778R2

PLOS One

Dear Dr. Kobayashi,

I'm pleased to inform you that your manuscript has been deemed suitable for publication in PLOS One. Congratulations! Your manuscript is now being handed over to our production team.

Kind regards,

on behalf of

Dr. Tadashi Ito

Academic Editor

PLOS One